# Palliative Care for Patients with Kidney Disease

**DOI:** 10.3390/jcm11133923

**Published:** 2022-07-05

**Authors:** Iacopo Lanini, Sara Samoni, Faeq Husain-Syed, Sergio Fabbri, Filippo Canzani, Andrea Messeri, Rocco Domenico Mediati, Zaccaria Ricci, Stefano Romagnoli, Gianluca Villa

**Affiliations:** 1Department of Health Sciences, Section of Anesthesia, Intensive Care and Pain Medicine, University of Florence, 50121 Florence, Italy; iacopo.lanini@unifi.it (I.L.); zaccaria.ricci@unifi.it (Z.R.); stefano.romagnoli@unifi.it (S.R.); gianluca.villa@unifi.it (G.V.); 2Department of Nephrology and Dialysis, ASST Lariana, S. Anna Hospital, 22042 Como, Italy; sarasamoni1@gmail.com; 3Department of Internal Medicine II, Division of Nephrology, Pulmonology and Critical Care Medicine, University Hospital Giessen and Marburg, 35392 Giessen, Germany; faeq.husain-syed@innere.med.uni-giessen.de; 4Department of Anesthesia and Intensive Care, Section of Oncological Anesthesia, and Intensive Care, Azienda Ospedaliero Universitaria Careggi, 50134 Florence, Italy; 5Palliative Care, USL Toscana Centro, 50122 Florence, Italy; filippo.canzani@uslcentro.toscana.it (F.C.); andrea.messeri@uslcentro.toscana.it (A.M.); 6Department of Anesthesia and Intensive Care, Section of Pain Therapy and Palliative Care, Azienda Ospedaliero Universitaria Careggi, 50134 Florence, Italy; roccodomenico.mediati@uslcentro.toscana.it; 7Pediatric Intensive Care Unit, Meyer Children’s University Hospital, 50139 Florence, Italy

**Keywords:** palliative care, end of-life care, chronic kidney disease, end-stage kidney disease, acute kidney injury, kidney replacement therapy

## Abstract

Interest in palliative care has increased in recent times, particularly in its multidisciplinary approach developed to meet the needs of patients with a life-threatening disease and their families. Although the modern concept of palliative simultaneous care postulates the adoption of these qualitative treatments early on during the life-threatening disease (and potentially just after the diagnosis), palliative care is still reserved for patients at the end of their life in most of the clinical realities, and thus is consequently mistaken for hospice care. Patients with acute or chronic kidney disease (CKD) usually experience poor quality of life and decreased survival expectancy and thus may benefit from palliative care. Palliative care requires close collaboration among multiple health care providers, patients, and their families to share the diagnosis, prognosis, realistic treatment goals, and treatment decisions. Several approaches, such as conservative management, extracorporeal, and peritoneal palliative dialysis, can be attempted to globally meet the needs of patients with kidney disease (e.g., physical, social, psychological, or spiritual needs). Particularly for frail patients, pharmacologic management or peritoneal dialysis may be more appropriate than extracorporeal treatment. Extracorporeal dialysis treatment may be disproportionate in these patients and associated with a high burden of symptoms correlated with this invasive procedure. For those patients undergoing extracorporeal dialysis, individualized goal setting and a broader concept of adequacy should be considered as the foundations of extracorporeal palliative dialysis. Interestingly, little evidence is available on palliative and end of life care for acute kidney injury (AKI) patients. In this review, the main variables influencing medical decision-making about palliative care in patients with kidney disease are described, as well as the different approaches that can fulfill the needs of patients with CKD and AKI.

## 1. Introduction

Palliative care includes multidisciplinary strategies aimed at supporting clinicians in managing patients with serious illnesses [1,2]. This innovative perspective has been developed to recognize and meet the needs of patients with a life-threatening disease and their families. It is mainly based on a multidimensional analysis including the identification and management of patients’ physical and psychological symptoms, social and spiritual needs, the assessment of patients’ clinical conditions and prognosis to establish realistic and appropriate treatment goals, the preparation of individualized treatment plans according to patients’ wishes, attention to families’ needs, and support for health care providers [2].

In current practice, patients considered as the most suitable for palliative care are those for whom curative treatments have been declared to have failed [1,3]. Consequently, most healthcare providers identify palliative care as a synonym for end-of-life care and its timing of initiation with the interruption of life-prolonging treatments [3]. Nevertheless, the limitation of palliative care to the very late phase of life may fail to support patients’ physical and psychological symptoms during the entire course of their disease [3,4]. Nowadays, the simultaneous provision of palliative care and life-sustaining treatments is recommended from the diagnosis of a serious disease (such as cancer and chronic organ dysfunction), and it should be considered within a comprehensive strategy for both critically and non-critically ill patients. In this model of “simultaneous care”, palliative care is neither an exclusive alternative to intensive curative treatments nor a follow-up to failed attempts to prolong patients’ survival [1,2]. The simultaneous care model could basically be offered to all patients with a life-threatening condition, regardless of factors such as age, comorbidities, or frailty. Instead, according to the definition developed by the Center to Advance Palliative Care and the American Cancer Society, “palliative care is appropriate at any age, at any stage of a serious illness, and may be provided in conjunction with curative treatment” [1,3]. On the other side, end-of-life care should be applied for patients with an expected survival of less then 6 months and who undergo restorative treatments and forego curative therapies [1,3].

Patients with acute kidney injury (AKI), chronic kidney disease (CKD), and particularly with end-stage kidney disease (ESKD) have a shorter life compared to those without kidney disease, despite the modern advances in therapeutic strategies. Extracorporeal kidney replacement therapies (KRTs) can increase the survival rates of these patients [1]. Nevertheless, these procedures may fail to improve outcomes in specific subgroups of acute, critically ill end-of-life patients, inappropriately prolonging and worsening the dying process of these patients [1].

Several clinical studies have demonstrated the benefits of proactive adoption of palliative care in patients with serious illness, e.g., cancer, some neurodegenerative diseases, and chronic organ dysfunction [1,5,6,7,8] (Table 1). Nevertheless, among patients with organ dysfunction, those with CKD or AKI rarely benefit from palliative care according to current practice, and mostly with a heterogeneous approach across different countries. Currently, the adoption of palliative care in the population with kidney disease is recognized mainly in the United Kingdom, the United States, Italy, and Canada for patients with ESKD [1,9,10,11,12,13]. Unfortunately, it is confined to patients with end-stage organ disfunction and very close to death. Furthermore, inadequate programs are still lacking for patients with AKI.

This review aims to analyze the variables influencing decision-making about palliative care and the approaches towards improving the quality of care in patients with renal diseases [1].

## 2. Palliative Care in Patients with Advanced CKD

The demand for dialysis steadily increases worldwide, particularly among elderly patients [14,15]. As demonstrated by the European Renal Association-European Dialysis and Transplant Association (ERA-EDTA), extracorporeal dialysis has doubled from 1980 to 2005, being applied to 55% of patients aged at 65 years with ESKD [1,16]. Most of these patients are treated with KRT three times per week in an outpatient dialysis clinic, requiring frequent travel that can be particularly challenging for elderly and frail patients [17].

Technological advances characterizing the maintenance of hemodialysis made these techniques largely effective in prolonging life expectancy during ESKD [1]. Nonetheless, the high prevalence of other non-renal chronic diseases (mainly metabolic and/or cardiovascular comorbidities) leads to a high mortality rate for ESKD patients undergoing maintenance hemodialysis, still quantified at about 23% per year [1,18,19]. Furthermore, a worsening in global functional status during the first year after hemodialysis initiation is commonly observed [1,20], as well as the worsening of physical, psychological, and social conditions [1,21]. The burden of symptoms led by the maintenance hemodialysis is comparable to that led by chemotherapy for advanced cancer [1,22]. Palliative care should thus be considered for these patients.

Data in the literature demonstrate how a proactive and early integration of palliative care in the treatment of CKD patients improves patient outcomes [23]. Interestingly, only a multi-professional team can fully meet the needs of the patient, especially in these highly complex conditions. All physicians caring for patients with advanced CKD, as a whole team, are encouraged to proactively promote discussion with patients and their families. The responsibility for the decision, timing, and contexts in which to initiate the discussion on renal maintenance replacement or palliative care lies with these healthcare professionals, according to their area of expertise and experience. In particular, the nephrologist is familiar with the trajectory of renal disease, and the physician deals with primary needs and palliative care, knowing the economic, family, social and psychological context of patients and their families.

The dynamic adjustment of KRT and the constant rediscussing of its targets is not the only approach to fulfill the patient’s needs to improve quality of life. After discussion and agreement with caregivers, a high percentage of patients regret starting extracorporeal treatments and, therefore, prefer conservative management of ESKD [1,24]. Education, information, and support for patients, their families, and caregivers are essential in planning the proper care management of patients with ESKD. Other treatment options, such as peritoneal dialysis (PD) and conservative therapy (see below), should be discussed and shared comprehensively among patients, families, and caregivers. For patients undergoing KRT and those managed with conservative therapy, palliative medicine should be considered to improve their quality of life [1].

### 2.1. Hemodialysis

Hemodialysis is a treatment widely used to replace kidney function during ESKD [25]. The appropriateness of the solute clearances in extracorporeal treatments is usually quantified through the measurement of Kt/V, i.e., the urea clearance normalized to total body urea [1]. Historically, given the relationship between Kt/V and mortality, this measure has been used to guide the customization of prescribed treatments. However, the solid effort for personalizing therapies has also promoted a careful evaluation of specific goals, which should aim not only at removing solute but rather at improving the patient’s entire clinical picture. Although reasonable, this concept is still far from being standard practice in most dialytic centers [1]. Indeed, the Dialysis Outcomes and Practice Patterns Study (DOPPS) has not shown any difference in dialysis prescription among patients with different clinical needs [26]. In addition to these specific clinical issues, the concept of hemodialysis adequacy should also consider achieving other treatment goals, such as improving the quality of life of patients and their families, preventing, and relieving suffering, and identifying and treating pain and other physical and psychosocial conditions. The role of palliative care in ESKD patients undergoing hemodialysis is to help the patient decide what is appropriate for himself, far beyond the purifying capacity of extracorporeal treatment. The concept of palliative dialysis has been proposed for specific subgroups of patients as a change in perspective. Ultrafiltration rate or dialytic clearance should be precisely set, based on a tight multidimensional evaluation of the actual patients’ requirements [1,16]. In this setting, a personalized hemodialysis prescription fulfills the concept of palliative care, improving physical, emotional, and spiritual symptoms and patient autonomy (Figure 1), reducing the amount of time spent in the hemodialysis unit or the clinical consequences of disproportionately intense sessions.

Notably, palliative dialysis is not understood as something different from chronic dialysis. The aims of chronic dialysis change, as it does not only have KV/T and thus solute clearance as its sole objective, but also the needs of more genial patients who may also have nothing to do with clearance. Palliative dialysis is not different from chronic dialysis, it just has broader goals and tries to embrace patients’ needs in a multidimension approach (Figure 1).

For these reasons, palliative management should not be initiated at patients’ end-of-lives but in conjunction with hemodialysis [1,16].

The PEACE tool can be used to assess the quality of care provided; a simple instrument that addresses six domains on physical symptoms, emotional symptoms, patients’ autonomy-related symptoms, communication, financial and spiritual issues [27]. Notably, the autonomy of the patients with ESKD and their relatives providing support are probably the most crucial factors to consider for clinical decision-making [1,16]. Evidence in the literature suggests that 84% of caregivers reluctantly change their personal and professional routines, feeling overburdened by the situation [1,22].

Palliative dialysis can effectively manage most of the frequent symptoms observed in patients with ESKD, such as dyspnea, lack of energy, drowsiness, dry mouth, pain, sleep disturbances, restless legs, itchiness, dry skin, and constipation [1,16,28]. Other symptoms can be treated with medication. However, fluid overload and acidosis are rarely controlled in patients with ESKD managed conservatively [1,16].

### 2.2. Conservative Management

Conservative management is considered for those patients unsuitable for or reluctant to KRT. It includes pharmacological and behavioral approaches to prevent the main consequences of ESKD, e.g., hydro-electrolyte imbalance, acid-base disorders, azotemia, and anemia. Geriatric patients, particularly if affected by dementia and frailty, are those most often suitable for conservative management [1,16].

Frailty, specifically, is something multifaceted, representative of a decline in health and organ function frequently observed (although not exclusively) in the elderly; it is found in almost 67% of dialysis patients [1,29]. It increases the risk of disability, hospitalization, institutionalization, and death [30]. Involuntary weight loss, slow walking, weakness, exhaustion, and low level of physical activity may be reliable diagnostic markers [1,31]. Patients with this clinical picture may require more conservative management [1]. Certainly, the nephrologist has a key role to play in guiding the discussion on discontinuing or withdrawing extracorporeal depuration in favor of conservative management. However, general practitioners and palliative care specialists should be involved in the discussion, as key providers of primary and specialist palliative care.

In addition, some tools for patients’ multidimensional assessment have recently been proposed and validated. They consider different patients’ features, such as biological, functional, psychological, and social features that define frailty. Some of them have been demonstrated to predict mortality, disability, and hospitalization, supporting decisions in CKD elderly patients [32].

### 2.3. Peritoneal Dialysis

Peritoneal dialysis (PD) is another approach available for the management of kidney disease. It can offer advantages over both hemodialysis and conservative management. Hemodynamic instability and severe hypotension can hinder hemodialysis, worsening the clinical condition of frail patients [1]. PD is considered a less invasive treatment, providing slow and continuous solute clearance and net ultrafiltration, preserving better kidney functions, allowing less restricted and more manageable diets [1]. Frail patients generally tolerate PD better and can maintain a better quality of life. Like hemodialysis, PD can compensate for the metabolic acidosis and fluid overload that can cause the exacerbation of symptoms in conservative management [1]. However, PD can also be controlled by domiciliary care and occasionally scheduled outpatient visits, making it a compelling alternative option [1,33].

In 2008, the ERA-EDTA outlined criteria that could be used to recommend PD [1,34]. The conditions of frail patients, such as the complications of ageing, severe heart disease, and peripheral vascular disease, were recognized as potential indicators for its use. However, limiting factors have also been identified, including inadequate physical capacity and a lack of family or social support [1]. Indeed, family involvement is more challenging in PD than in hemodialysis and conservative management, considering that technical training of the patient and/or caregiver is mandatory [1]. Although close collaboration among patients, their families, the general practitioner, nurses, and the nephrologist are necessary, PD may allow a better quality of life, thus making it a therapy of choice in a selected population [1].

Maximum conservative management (MCM) constitutes, in this perspective, a European multidisciplinary proposal with various healthcare professionals with the aim of improving the quality of life of frail ESKD patients [1,35], not necessarily prolonging survival [1,36]. The MCM proposal is multidisciplinary, aimed at improving the quality of life, not prolonging it, the need for superior home care, adaptation of treatment to avoid ER visits, procedures, hospitalizations, the need for communication between patient, family, providers (general and specialist). Although it needs less institutionalization, the conservative approach requires multiprofessional superior home care and constant adaptation of treatment to avoid failure, emergency room admissions, invasive procedures, and hospitalizations. A constant sharing of achievable treatment goals between specialists, general practitioners, patients, and family members should be encouraged [1].

## 3. Palliative Care for AKI Patients

The overall prevalence of AKI cases developed among critically ill patients in ICUs is between 36% and 67% [1,37]. The pathophysiological mechanisms and etiologies may be multiple, and the mortality rate is still conspicuous [1,38,39,40,41,42]. Indeed, severe AKI is characterized by a poor prognosis, especially given the critical illness, multiorgan dysfunction, and the need for extracorporeal renal support, and the concepts of palliative and end-of-life care are still poorly explored in these cases [1]. In fact, few works have appeared in the scientific literature on the withdrawal or discontinuation of invasive treatments in cases of AKI, despite the growing buzz about palliative care in critically ill patients [1]. A systematic literature search published in PubMed in 2015, with the following keywords ‘acute kidney injury’ AND ‘palliative care’ OR ‘palliative medicine’ and related MeSH terms, reported only 102 citations [1]. Of these, 15 citations were focused on ‘acute-on-chronic’ conditions and advanced planning for ESKD cases. Palliative care for patients with multiple organ dysfunction (when AKI is associated with other serious conditions, e.g., heart failure or cancer) had been explored in 32 papers [1]. Only 12 articles were focused explicitly on adoption of palliative care among AKI patients. Five articles reviewed ethical issues [43,44,45,46,47] and three articles described the epidemiology and clinical conditions during the end of life of AKI patients [1,48,49,50]. Notably, the same literature search performed nowadays using the same research questions (June 2022), would identify 128 papers. Only twenty-seven articles have been added in PubMed on this topic in the last seven years.

Thus, although palliative and hospice care management are considered comprehensively in individuals with CKD, there is still a lack of general guidance for those with acute illness and AKI. In recent times, physicians frequently encounter critically ill patients who meet the criteria for KRT initiation; however, strong doubts remain about the real benefit of this practice [1,51]. However, mortality in intensive care remains very high, despite the advancement of targeted diagnostic and therapeutic interventions [1,52]. Advanced life-support systems, such as acute KRT, are an option even though they can only prolong the dying process unreasonably [1,53]. Similar ethical debacles can be found in the literature for other life support systems, such as mechanical ventilation, extracorporeal membrane oxygenation or extracorporeal CO2 removal. Several instruments have been proposed in the literature with prognostic intentions and with the aim of supporting physicians in guiding extracorporeal treatments. Among others, other tools have been proposed to recognize patients at the end of life for whom discontinuation or elimination of treatments might be appropriate [54].

Little evidence is available on the discontinuation of KRT in cases of lack of future benefits [1,51]. Although the Renal Physicians Association and the American Society of Nephrology have already created an ad hoc line [46], only a few nephrologists and intensivists are aware of it [1,12]. Therefore, withholding or discontinuation of acute KRT may be based primarily on local institutional practice, clinical judgement, available resources, and local management [40,42,55,56,57].

During the decision-making process, several aspects must be considered regarding restraint or interruption of KRT in AKI cases, including feasibility, prediction of survival, prediction of renal functional recovery, and quality of life [1].

Clinical feasibility is undoubtedly a significant limiting factor. Although the clinical conditions of several patients, including those with severe hypotension, might adversely affect technical feasibility, certain techniques, such as continuous renal replacement therapies, have made it possible to perform KRT on most patients [51].

Appropriate medical judgment supported by informed consent cannot disregard survival prediction as a key element in a decision to continue, discontinue or withdraw KRT. Despite numerous organ dysfunction scoring systems [1,58] and available outcome prediction models, none of them provides sufficient information on the appropriateness of KRT for the individual patient [1,53]. Short- and long-term mortality rates in AKI and KRT cases are high globally (46–75%) [1,59]. The Study to Understand Prognoses and Preferences for Outcomes and Risks of Treatments (SUPPORT), the largest prospective study on critical care survival, revealed that the median survival in patients who have undergone dialysis was about 30 days; more interestingly, only one third of them were still alive after 5 months [1,60].

In a decision-making process [50], for the purpose of determining long-term renal and non-renal outcomes [61,62,63,64], a prediction of renal recovery after AKI must also be evaluated along with other factors [1]. Patients’ and their families’ quality of life may be severely deteriorated during ESKD and if maintenance dialysis is required post AKI. However, long-term outcomes and quality of life in critically ill patients requiring KRT have been little investigated in the literature [1,51]. In the SUPPORT study, for example, surviving critically ill patients with AKI manifested an average of one dependency in activities of daily living [58], although various studies publish different results [1,47,65,66].

In contrast to CKD cases, there is often not enough time to decide to start or stop KRT in critically ill AKI patients. The very fact of sometimes not being aware of the patient’s wishes, and thus making a clinical judgement even more complex [67], results in an accumulation of decision-making pressure on the part of the family and the healthcare team [1,51]. According to the Renal Physicians Association/American Society of Nephrology guidelines on shared decision-making in dialysis, a time-limited trial of KRT could be considered for patients with uncertain prognoses [1,68]. Time-limited trials can be useful in managing disagreement between physicians or nurses and patients’ families [1,51]. The endpoints, objectives and duration of this time-limited trial must be declared in advance. The specific criteria, the magnitude of change accepted as evidence of improvement and the time of evaluation need to be agreed upon between physicians, nurses, patients, and their relatives [1,51]. In particular, the decision-making process regarding interruption or discontinuation of KRT in AKI cases in this time-limited trial is a continuous process: clinical outcomes and prognoses need to be reassessed as needed [1,51].

In all those situations where KRT is futile, the physician should provide adequate comfort care for those critically ill patients with kidney disease at the end of their life [1,69]. Notably, “futile” is the definition used for treatments that do not provide the patient’s desired outcomes or that do not achieve the doctor’s goal of making the patient stable enough to be transferred to a less intensive level of care. Treatments are thus practically decided by the physician basing his decision on what are the realistic prospects of cure of the therapy delivered to a specific patient with his own expectations and personal preferences. This qualitative approach, hind at fulfilling physical and not physical patients’ needs, typically excludes extracorporeal treatments, preferring instead pharmacological or non-pharmacological, capable of alleviating patients’ global suffering [1]. Unfortunately, unlike CKD patients, critically ill patients with AKI very uncommonly have the possibility to declare expectations and preferences due to delirium or unconsciousness. Many countries have adopted advance treatment provisions that may help in these clinical conditions. Unfortunately, these advanced provisions are rarely available in most of the patients, particularly for those for whom critical illness was unexpected and developed quickly. In these cases, family members are asked about significant communications on patients’ values, preferences, expectations that took place before their admission to the ICU. These “reported provisions” are used by physicians to adjust treatments. Nowadays, the definition of patient-centered futility in the ICU patient remains an open question.

In cases with a prognosis of death, the use of approaches other than integrated palliative care and intensive care would be desirable. Indeed, hospice can provide these in such situations [1,2]. In terminally ill patients, strategies to ensure a good death often go beyond adequate analgesia. Theoretically, palliative care should aim to optimize the patient’s comfort and dignity, for example by allowing a death within the patient’s own home [70], or by providing support to the family [1]. Unfortunately, the dying process is prolonged inappropriately in hospitals; patients may receive unwanted, expensive, and invasive treatments that can generate further discomfort, such as pain, dyspnea, thirst, and anxiety [1,71].

New technologies, such as the wearable artificial kidney, should be evaluated about hospice care during renal problems. Although difficult to apply in daily clinical practice, it may prospectively represent a useful tool in the home care of patients with AKI. This miniaturized, wearable technology can provide mechanical support, mainly through continuous ultrafiltration. By ameliorating fluid overload and reducing dyspnea, the wearable artificial kidney can thus support basic human needs at the patient’s end of life at home [1,65].

Finally, the concepts of palliative KRT that may apply for chronic ESKD patients could theoretically be considered even for CRRT patients, for example, with fewer intensive treatments aiming to avoid/prevent pulmonary edema in a extubated patient or severe electrolyte/acid-base derangements without the aim of specific solute control or a dialytic dose target.

## 4. Conclusions

Palliative care is mostly directed at all cases of patients with serious illness in the terminal phase of the disease and is often confused with hospice care. Early recourse to palliative care for all the above-mentioned cases can enhance the quality of life and support families at the same time. Renal patients generally have a shortened life expectancy, thus being able to benefit from palliative care, the latter being a well-defined concept especially for CKD rather than AKI. Indeed, different therapeutic strategies can be pursued to improve the quality of life in CKD patients, including palliative dialysis, conservative management, and PD. All these procedures require close collaboration between healthcare professionals, patients, and families. Prognosis, realistic treatment goals and therapeutic decisions must be shared by all parties. There are few and inadequate data in the literature on the effects of palliative and hospice care in AKI patients. A methodological, ethical, and clinical effort is needed to implement knowledge and awareness of this under-explored topic.

## Figures and Tables

**Figure 1 jcm-11-03923-f001:**
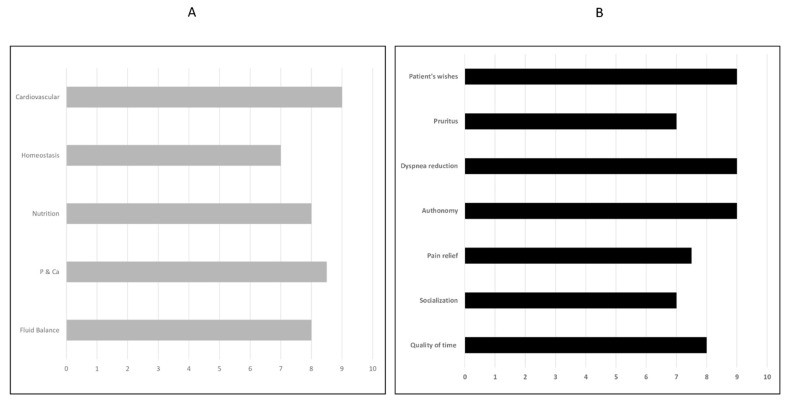
“Adequacy” of extracorporeal treatment and the role of palliative dialysis. Graph of a chronic extracorporeal renal substitution for CKD patient (panel **A**) and graph of palliative dialysis for “CKD end-of-life” patient (panel **B**). Each graph considers several variables differently affected by the treatments (0: the variable is minimally influenced; 10: the variable is strongly improved). The area may identify the adequacy of the treatment within the graph: during an ideal therapy, the inner area covers 100% of the graph. For a treatment to be “adequate”, other parameters than solute clearance or fluid balances should be considered for end-of-life patients; in this scenario, an individualized RRT prescription should improve the physical, emotive, and autonomy-related issues.

**Table 1 jcm-11-03923-t001:** AKI and CKD patients’ needs and strategies.

	Patients’ Needs	Strategies
Acute kidney injury	Clearance of uremic solutes for cognitive impairment pH homeostasis for dyspnea Fluid removal for dyspnea	Acute KRT
Chronic kidney disease	Clearance of uremic solutes and pruritus pH homeostasis for dyspnea Fluid removal for dyspnea Control of physical symptoms	Maintenance KRT
	Autonomy and self determination Maintenance of social activities Clearance of uremic solutes and pruritus	Peritoneal dialysis
	Autonomy and self determination Prevention of social/psychological/physical distress	Conservative treatment

## Data Availability

Not applicable.

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
