# Peer review of "Palliative Care for Patients with Kidney Disease"

_jcm, 2022, doi:10.3390/jcm11133923_

Round 1
Reviewer 1 Report
This is a manuscript regarding to palliative care for patients with kidney disease. The manuscript is interesting, especially for palliative care for patients with acute kidney disease. I think it's a theme I've never seen so far summarized. Minor comments
1. I think it will be easier to see if the results can be summarized in a table.
Author Response
Table created and added as a file also in the text.

Reviewer 2 Report
In their review article on "Palliative care for patients with kidney disease," Lanini and colleagues discuss the literature on applying a multidisciplinary approach to caring for patients with chronic kidney disease, their families and support the primary team as it deals with the physical and psychological aspects of the patient's illness. They note that palliative care is more than steps for patients at the end of life, for whom curative treatment has failed. They break the discussion into sections on palliative care in advanced CKD, in acute kidney injury with CKD, in hemodialysis, in peritoneal dialysis and in end stage disease that is treated supportively without dialysis. They discuss the role of palliative care in discussions of removing dialysis treatment when it is futile and cite the relevant literature from Europe and the US. The key point they make is that palliative care is something that should be engaged even while curative treatment is being undertaken, not just at the end of life.
The organization of the paper takes the reader through the different stages of chronic kidney disease, the treatment modalities for end stage renal disease and the approach to conservative care in end stage disease without dialysis. It also looks at the role of palliative care in the acute setting, which renal replacement therapy must be offered quickly and where it is life support rather than maintenance therapy.
There are several major criticisms that I think the authors must address. First, where and when should discussions about maintenance renal replacement take place - with the primary nephrologist, with the primary care physician, with both?
To whom should simultaneous palliative care be offered. Just older patients whose life expectancy on dialysis is short, due to age, co-morbidities, frailty, to anyone for whom a poor outcome is expected on dialysis.
Likewise, who should conduct the discussion about conservative renal care in end stage disease - the primary care physician, the nephrologist, the palliative care specialist?
Is there any data that engaging palliative care early improves outcomes for patients with CKD progressing to end stage disease. Presumably these are outcomes set by the patient
In the section on hemodialysis, the authors do an admirable job in discussing the inadequacy of the definition of adequacy as nothing more than small molecule solute clearance. They should define KT/V in terms of urea clearance normalized to total body urea and explain why this has come to be defined as adequacy (mortality). The authors should make the point more forcefully the the role of palliative care in dialysis should be in helping the patient decide what adequacy is beyond a unitless number.
The discussion of palliative dialysis would be better in a separate section on helping patients and families stop chronic dialysis. Table 1 could be used to highlight the discussion, showing how a prescription for palliative dialysis adequacy differs from a standard dialysis prescription for adequacy.
Discussion of removing patients from dialysis in the acute setting would be better folded into a section on the role for palliative care in helping patients with acute kidney injury requiring dialysis. Indeed the authors should address whether the decision to provide dialysis under those circumstance would be better served in a discussion on invasive life sustaining measures like CPR, intubation, vasopressors.
The paragraph starting on line 171 "Maximum conservative management..." needs more specifics about what the MCM proposal includes or tighter structure to highlight the main elements: multidisciplinary, aimed at improving quality of life, not prolonging it, need for superior home care, adaptation of treatment to avoid ER visits, procedures, hospitalizations, need for communication between patient, family, providers (general and specialist). I would also put this section after the section on PD.
At line 275, the authors need to define futile, which is a loaded term in discussion about ending care. How is it defined - therapy that does not provide the outcome desired by the patient or therapy that will not achieve the physicians goal of making the patient stage enough to transfer to a less intensive level of care. Who decides what it is futile? What is a patient centered way to define futile in stopping dialysis, acutely or chronically?
The authors seem to be getting at this in the sentence at line 267 "Disagreement in management between physicians or nurses and patients' families me be helpful." In what way and to whom?
Author Response
There are several major criticisms that I think the authors must address. First, where, and when should discussions about maintenance renal replacement take place - with the primary nephrologist, with the primary care physician, with both?
We strongly believe that a multidisciplinary approach is the only one that may guarantee successful palliative care, and thus it should be encouraged. Thus, the responsibility for the decision, timing, and settings for initiating discussion on maintenance renal replacement belongs to all physicians who take care of the patients. In particular, the nephrologist is familiar with the trajectory of the kidney disease, while the physician deals with basic needs and palliative care, knowing the economic, family, social, and psychological environments of the patients and their families. Added in the text on line 113.
To whom should simultaneous palliative care be offered? Just older patients whose life expectancy on dialysis is short, due to age, co-morbidities, frailty, to anyone for whom a poor outcome is expected on dialysis?
Concurrent palliative care could be offered basically to all those patients with a life-threatening condition regardless of factors such as age, co-morbidities, frailty. Added in the text on line 68.
Likewise, who should conduct the discussion about conservative renal care in end stage disease - the primary care physician, the nephrologist, the palliative care specialist?
The discussion on renal-conservative care in end-stage disease should involve all the specialists mentioned. The scientific explanations should be given by the nephrologist, but then needs the physician and the palliative therapist to fulfill basic and specialist palliative care (line 198).
Is there any data that engaging palliative care early improves outcomes for patients with CKD progressing to end stage disease? Presumably these are outcomes set by the patient.
This issue has been included in the main discussion.
In the section on hemodialysis, the authors do an admirable job in discussing the inadequacy of the definition of adequacy as nothing more than small molecule solute clearance. They should define KT/V in terms of urea clearance normalized to total body urea and explain why this has come to be defined as adequacy (mortality). The authors should make the point more forcefully the role of palliative care in dialysis should be in helping the patient decide what adequacy is beyond a unitless number.
We agree with the reviewer's comment. We have modified the main text as required (line 150).
The discussion of palliative dialysis would be better in a separate section on helping patients and families stop chronic dialysis.
In our paper, palliative dialysis is not intended to be as something different from chronic dialysis. The purposes of chronic dialysis change, not only having KV/T and thus solute clearance as the sole objective, but also the needs of patients who are more genial and who may also have nothing to do with clearance. Palliative dialysis is no different from chronic dialysis, it just has broadened goals and tries to embrace patients' needs in a multidimensional way (line 166). Figure 1 could be used to highlight the discussion, showing how a prescription for palliative dialysis adequacy differs from a standard dialysis prescription for adequacy. As explained above.
Discussion of removing patients from dialysis in the acute setting would be better folded into a section on the role for palliative care in helping patients with acute kidney injury requiring dialysis. Indeed, the authors should address whether the decision to provide dialysis under those circumstance would be better served in a discussion on invasive life sustaining measures like CPR, intubation, vasopressors.
We agree with the reviewer's comment. We have modified the main text as required. (line 265).
The paragraph starting on line 171 "Maximum conservative management..." needs more specifics about what the MCM proposal includes or tighter structure to highlight the main elements: multidisciplinary, aimed at improving quality of life, not prolonging it, need for superior home care, adaptation of treatment to avoid ER visits, procedures, hospitalizations, need for communication between patient, family, providers (general and specialist). I would also put this section after the section on PD.
Done in the text on line 248.
At line 275, the authors need to define futile, which is a loaded term in discussion about ending care. How is it defined - therapy that does not provide the outcome desired by the patient or therapy that will not achieve the physicians goal of making the patient stage enough to transfer to a less intensive level of care. Who decides what it is futile?
Definition of futile has been provided in line 349 and the discussion implemented as suggested.
What is a patient centered way to define futile in stopping dialysis, acutely or chronically?
The discussion has been implemented with these issues (line 356). It has been expressed that different from CKD patients, critically ill patients with AKI very uncommonly have the possibility to declare expectations and preferences due to delirium or unconsciousness. Many countries have adopted advance treatment provisions that may help in these clinical conditions. Unfortunately, these advanced provisions are rarely available in most of the patients, particularly for those for whom critical illness has been unexpected and promptly developed. In these cases, cases the family members are asked about significant communications on patient’s values, preferences, expectations that took place before admission to the ICU. These “reported provisions” are used by physicians to adjust treatments. Nowadays, the definition of patient-centered futility in the ICU patient remains an open question.
The authors seem to be getting at this in the sentence at line 267 "Disagreement in management between physicians or nurses and patients' families me be helpful." In what way and to whom?
“Time-limited trials can be useful in managing disagreement between physicians or nurses and patients' families” is the right phrase. Sorry for the mistake. Modified in the text on line 335.
This manuscript is a resubmission of an earlier submission. The following is a list of the peer review reports and author responses from that submission.